# Prophylactic Effects of Bee Venom Phospholipase A2 in Lipopolysaccharide-Induced Pregnancy Loss

**DOI:** 10.3390/toxins11070404

**Published:** 2019-07-12

**Authors:** Hyunjung Baek, HyeJin Yang, Jong Hoon Lee, Na-Hoon Kang, Jinwook Lee, Hyunsu Bae, Deok-Sang Hwang

**Affiliations:** 1Department of Physiology, College of Korean Medicine, Kyung Hee University, Seoul 02453, Korea; 2Department of Clinical Korean Medicine, Graduate School, Kyung Hee University, Seoul 02453, Korea

**Keywords:** bee venom phospholipase A2, spontaneous abortion, Tregs, pregnancy, depletion of regulatory T cell (DEREG) mouse model

## Abstract

Spontaneous abortion represents a common form of embryonic loss caused by early pregnancy failure. In the present study, we investigated the prophylactic effects of bee venom phospholipase A2 (bvPLA2), a regulatory T cell (Treg) inducer, on a lipopolysaccharide (LPS)-induced abortion mouse model. Fetal loss, including viable implants, the fetal resorption rate, and the fetal weight, were measured after LPS and bvPLA2 treatment. The levels of serum and tissue inflammatory cytokines were determined. To investigate the involvement of the Treg population in bvPLA2-mediated protection against fetal loss, the effect of Treg depletion was evaluated following bvPLA2 and LPS treatment. The results clearly revealed that bvPLA2 can prevent fetal loss accompanied by growth restriction in the remaining viable fetus. When the LPS-induced abortion mice were treated with bvPLA2, Treg cells were significantly increased compared with those in the non-pregnant, PBS, and LPS groups. After LPS injection, the levels of proinflammatory cytokines were markedly increased compared with those in the PBS mouse group, while bvPLA2 treatment showed significantly decreased TNF-α and IFN-γ expression compared with that in the LPS group. The protective effects of bvPLA2 treatment were not detected in Treg-depleted abortion-prone mice. These findings suggest that bvPLA2 has protective effects in the LPS-induced abortion mouse model by regulating Treg populations.

## 1. Introduction

Pregnancy has been considered a state of maternal immunological tolerance because the maternal immune system protects the semi-allogenic fetus during the implantation period. The feto–maternal interface creates a pregnancy-supportive immune environment and provides protection from harmful pathogens. The balance between immune tolerance and immune activation is important for the maintenance of pregnancy, and aberrant patterns are associated with complications such as implantation failure, pregnancy loss, and preterm labor [1]. The maternal immune response is regulated by cytokines acting on the Th1/Th2 balance to orchestrate the changes in leukocyte populations required to protect the conceptus during pregnancy [2,3] and facilitate the tissue remodeling processes necessary for adequate placental development [4]. 

Spontaneous abortion, also known as miscarriage, is the most common complication of early pregnancy and is classically defined as the natural death of an embryo or fetus before 20 weeks of gestation without a surgical procedure [5,6]. It is estimated that 10–15% of women who conceive experience one or more miscarriages, and an estimated 1~3% of women suffer from recurrent pregnancy loss (RPL) [7]. Spontaneous abortion has multiple etiologies, including maternal immune incompetence, chromosomal anomalies, infections, and hormonal insufficiency [8,9]. It is well known that the two main mechanisms involved in abortion are the inflammatory process and thrombosis [10]. An increased inflammatory response in the decidua is frequently reported in women who develop RPL. The inflammatory cytokine level in the placental–decidual interface is a key determinant of the viability and development potential of a fetus. Therefore, adequate immune adaptations play a central role in the prevention of pregnancy disorders, including recurrent pregnancy loss, preeclampsia, fetal growth restriction, and premature birth [11]. 

Regulatory T cells (Tregs) are known as key regulators that maintain tolerance of self-antigens and prevent autoimmune disease. Treg cells allow fetal survival by acquiring a transient state of tolerance for paternal alloantigens in fetal tissues during pregnancy. Mounting evidence has indicated that a deficiency in the Treg cell number and function is associated with unexplained infertility, abortion, and preeclampsia. In recent studies, peripheral and decidual Tregs were demonstrated to increase and suppress the responses to paternal alloantigens during pregnancy in humans and mice [12]. Women with spontaneous abortion showed decreased Tregs in the peripheral blood and decidua compared with those in an induced abortion group. Zenclussen et al. reported that normal pregnant mice showed higher frequencies of CD4^+^CD25^+^ and IL-10^+^ Treg cells in the thymus than abortion mice. Furthermore, in vivo prevention of the fetal resorption of abortion-prone mice was achieved after adoptive transfer of Treg cells from the spleens of normal pregnant mice at mid-pregnancy [13]. By contrast, the depletion of Tregs before mating significantly influences the uterine environment and leads to gestation failure [14]. 

Bee venom (BV) is a natural toxin produced by the honey bee and has commonly been used to treat arthritis and rheumatoid patients [15]. The enzyme bee venom phospholipase A2 (bvPLA2), one of the major components of BV, hydrolyzes membrane phospholipids to generate lysophospholipids and arachidonic acid. Previous findings from our group showed that the population of Treg cells was significantly increased by bvPLA2 treatment in vivo and in vitro [16]. bvPLA2 directly binds to CD206, a mannose receptor expressed on dendritic cells, and modulates the secretion of PGE2, resulting in Treg differentiation via PGE2-EP2 signaling [17]. In the present study, we investigated the prophylactic effects of bvPLA2 in the uterus using an LPS-induced model of abortion in pregnant mice. bvPLA2 pretreatment prevented fetal loss accompanied by growth restriction by inducing Treg cells. The production of proinflammatory cytokines, especially that of TNF-α, in serum and uterine tissue was elevated following LPS injection but was dramatically decreased after bvPLA2 treatment. Finally, we determined the effects of Treg depletion following bvPLA2 treatment in LPS-induced abortion. Treatment of LPS-induced abortion mice with bvPLA2 diminished the risk of fetal death. However, these protective effects were not detected in Treg-depleted abortion-prone mice. These findings suggest that bvPLA2 treatment is an important regulator of Treg-dependent maintenance of pregnancy.

## 2. Results

### 2.1. Effect of bvPLA2 Pretreatment on Fetal Loss in LPS-Induced Abortion Mice

To examine the effect of bvPLA2 on the susceptibility to LPS-induced fetal resorption, C57BL/6 female mice were mated with males and then were administered vehicle or LPS on gestation day (GD) 9.5. To examine the prophylactic effects of bvPLA2 treatment, bvPLA2 was administered by intraperitoneal (i.p.) injection once a week for two weeks prior to mating. Resorption occurring in response to treatment with LPS on GD 9.5 was characterized by extensive hemorrhage and complete maceration of the uterine contents 24 h after LPS administration. Resorption was analyzed on a per-implantation-site basis, and the resorbed embryos were atrophied and necrotic (Figure 1A). The numbers of viable implants were significantly decreased in LPS-administered pregnant females (the LPS group) compared to those seen in PBS-treated pregnant females (the PBS group) (Figure 1B). Conversely, bvPLA2 administration showed significantly increased numbers of viable fetuses compared with LPS-treated controls (the LPS group). Pregnant animals in the LPS + bvPLA2 group appeared to contain viable and healthy fetuses. LPS injection on GD 9.5 resulted in a mouse spontaneous abortion model, with an increased fetal resorption rate of 44.4% ± 9.6% (Figure 1C). This embryo resorption induced by LPS could be reversed by bvPLA2 treatment, which produced an embryo resorption rate of 25%, although it was slightly higher than that of the control group (22.2% ± 4.8%). When mice were examined on GD 11.5, LPS administration resulted in significantly smaller fetal weights (67 ± 15 mg) than those in PBS-treated mice (242 ± 28 mg) (Figure 1D). By contrast, there was a significant reduction in fetal weight between the LPS group (67 ± 15 mg) and LPS + bvPLA2 group (373 ± 105 mg). No significant differences were detected on fetal loss between the PBS and bvPLA2 groups (Figure 1). These results taken together, pretreatment of pregnant mice with bvPLA2 decreased the LPS-induced resorption frequency considerably. Next, to evaluate the effect of bvPLA2 administration on Treg populations, we examined the levels in reproductive tissues (Figure 1E). The percentage of Foxp3^+^Treg cells was significantly increased after pregnancy (the PBS group) compared with that in non-pregnant mice (the NP group). As shown in Figure 1E, treatment with bvPLA2 was associated with augmentation in the proportion of CD45^+^Foxp3^+^Treg cells in uterine tissues compared with that in the LPS group. We suggest that bvPLA2-mediated induction of Foxp3^+^Treg cells in uterine tissues may affect to the prevention of pregnancy loss. We further investigated the effects of bvPLA2 on the inflammatory cytokine levels, including IFN-γ and TNF-α, in uterine tissues.

### 2.2. Effect of bvPLA2 on the Inflammatory Cytokine Levels in LPS-Induced Abortion Mice

Next, the serum cytokine levels of each group were analyzed using the CBA mouse Th1/Th2/Th17 Cytokine Kit (Figure 2A). The IL-4 concentrations in serum in the LPS and LPS + bvPLA2 groups showed no significant differences compared with those in the PBS group (Figure 2B). The proinflammatory cytokines IFN-γ and TNF-α play important roles in the pathogenesis of early embryo loss. Compared with the expression in the PBS group, the IFN-γ and TNF-α expression levels in serum samples were significantly increased in the LPS group (Figure 2C,D). Most significantly, bvPLA2 treatment resulted in a substantial decrease in IFN-γ and TNF-α levels in all mice compared with those in LPS-injected mice. However, there was no apparent change in the serum levels of IL-4 by bvPLA2 treatment. There was no significant change in the serum IL-17A expression after LPS injection (Figure 2E). However, the IL-17A concentration was decreased significantly in the LPS + bvPLA2 group compared with that in the LPS group. Elevated serum IL-10 levels, produced by Treg cells, were detected only in the LPS + bvPLA2 group, representing the induction of Treg cells by bvPLA2 treatment (Figure 2F). 

### 2.3. bvPLA2 Pretreatment Inhibits LPS-Induced TNF-α Production

Next, we asked whether the elevated TNF-α levels in serum samples would also occur in the uterine tissues of LPS-injected mice and whether bvPLA2 administration would result in altered expression of TNF-α. By RT-PCR and ELISA analysis, we found that LPS injection resulted in a significant upregulation of TNF-α mRNA and protein expression compared with that in the PBS group (Figure 3A,B). Of interest, pretreatment with bVPLA2 significantly repressed the LPS-induced elevation of the TNF-α level.

### 2.4. Effects of bvPLA2 on Various Immune Cell Types in Mice Prone to LPS-Induced Spontaneous Abortion

To investigate the effects of bvPLA2 on various immune cell types in the uterus of LPS-induced abortion mice, uterine natural killer cells (uNKs), macrophages, and T helper and cytotoxic T cell populations among monocytes isolated from uterine tissues were analyzed using flow cytometry. The number of CD3^+^ total T cells was increased in the LPS group compared with that in the PBS group (Figure 4A). CD4^+^ T helper cells were increased in the LPS + bvPLA2 group compared with those in the NP, PBS, and LPS groups (Figure 4B). Interestingly, CD8+ cytotoxic T cell populations were significantly increased by LPS injection, whereas bvPLA2 administration showed similar CD8^+^ levels to those of the PBS group (Figure 4C). uNK cells play a major role in maintaining normal pregnancy and are recruited to the feto–maternal interface. An over twofold increase in uNK cells was detected in normal pregnant females compared with that in non-pregnant controls (Figure 4D). LPS treatment showed a reduction in the number of uNK cells to a level similar to that in non-pregnant controls. The numbers were restored by bvPLA2 pretreatment. The F4/80^+^CD11c^+^CD206^+^ M2 macrophage populations were dramatically increased in the LPS group but decreased in the LPS + bvPLA2 group (Figure 4E). However, no significant differences among all groups were detected in F4/80^+^CD11c^+^CD206^−^ M1 macrophages (data not shown). 

### 2.5. Effects of bvPLA2 Treatment on uNK Cells in the Decidua of LPS-Induced Abortion Mice

During normal mouse pregnancy, increased numbers of uterine natural killer (uNK) cells promote physiological changes in the endometrial structure, contributing to early, post-implantation endometrial angiogenesis and spiral arterial modification. Mouse uNK cells are recognized as lymphocytes containing periodic acid–Schiff (PAS)-reactive cytoplasmic granules or *Dolichos biflorus* agglutinin (DBA) lectin [18,19]. We observed a significant decrease in PAS+ uNK cells on GD 11.5 in the uterus of the mice subjected to LPS-induced abortion compared with that in control mice (Figure 5, upper panel). Pretreatment with bvPLA2 showed a significant increase in both PAS^+^ and DBA lectin^+^ uNK cells in the decidua of LPS-induced abortion female mice. 

### 2.6. Effects of Treg Depletion on Fetal Loss in LPS-Induced Abortion Mice Exposed to bvPLA2

Depletion of regulatory T cell (DEREG) mice express a diphtheria toxin receptor (DTR)-eGFP transgene under the control of the Foxp3 promoter, allowing for the specific depletion of Treg by diphtheria toxin (DT) injection [20]. For Treg depletion, DEREG mice were administered 1 µg of DT for two consecutive days 2 weeks before mating and again 3 and 7 days after mating. DT injection into DEREG mice resulted in the transient depletion of Treg cells from the peripheral blood. In a previous report, Foxp3^+^ Treg depletion was prolonged for 4 days in mice treated with 1 µg of DT for two consecutive days [21]. To examine the effects of Treg depletion on susceptibility to LPS-induced fetal resorption, female DEREG mice were mated with males and then were administered PBS, LPS, or LPS with bvPLA2 on GD 9.5. DT was injected for two consecutive days two weeks prior to mating and twice during pregnancy. Tregs were depleted by DT injection (Figure 6A). Tregs were measured as CD45^+^Foxp3^+^ cell populations in the uterine tissues obtained from each group. As shown in Figure 1E, the Treg populations of the bvPLA2-treated group were significantly higher than those in the PBS and LPS groups. DT administration did not induce CD45^+^Foxp3^+^ cell populations in bvPLA2-treated abortion-prone mice (the d-LPS + bvPLA2 group) (Figure 6A). Next, we examined the effects of Treg depletion on the fetal weight, viable implants, and resorption rate in LPS-injected mice exposed to bvPLA2. When the mice were examined on GD 11.5, LPS treatment showed a significantly smaller fetal weight than that in PBS-treated mice. By contrast, a dramatic reduction in the fetal weight was found between the LPS group and LPS + bvPLA2 group (Figure 6B). As expected, the dramatic reduction in the fetal weight between the LPS group and LPS + bvPLA2 group was not detected in the DT-injected groups. The number of viable implants was twofold higher in the LPS + bvPLA2 group than in the LPS group (Figure 6C). The fetal resorption rate induced by LPS treatment was decreased by bvPLA2 administration. However, the protective effects of bvPLA2 on viable implants and the fetal resorption rates disappeared following Treg depletion (Figure 6C,D). These results suggest that the protective effects of bvPLA2 on the LPS-induced resorption frequency depend on the presence of Treg populations.

## 3. Discussion

The present study showed that prophylactic treatment with bvPLA2 has protective effects on fetal growth restriction in response to an inflammatory insult of LPS-induced abortion. bvPLA2 administration diminished the fetal death and serum proinflammatory cytokine levels in an LPS-induced abortion mouse model. Finally, after the depletion of Treg cells using DEREG mice following DT administration, the protective effects of bvPLA2 treatment were not detected. The findings suggest that bvPLA2 treatment is a pivotal determinant of susceptibility or resilience to LPS-induced fetal pathologies due to the central role of Tregs in suppressing inflammatory cytokine production in the implantation site.

Current therapies being considered for the prevention of spontaneous miscarriage and the treatment of recurrent miscarriage include progesterone and related steroids; supplementing a pregnancy with progestin often begins early in pregnancy and can continue through the first trimester and beyond [22,23]. Progesterone, an essential hormone in the process of reproduction, induces secretory changes in the inner lining of the uterus and is important for healthy embryo development. However, the use of progesterone during early pregnancy remains controversial. Information about the side effects to the mother or child using progesterone is lacking. The use of low-molecular-weight heparin, aspirin, or both is also controversial, and the anti-miscarriage effect is not seen in women with thrombophilias [24,25]. In a recent report, sildenafil citrate (Viagra), either alone or in combination with heparin, was suggested as a clinically applicable therapy for patients with early pregnancy loss or recurrent miscarriages [26]. More efficient and safe treatment for patients who experience early pregnancy loss is urgently needed. We determined the modulation of fetal loss by bvPLA2 treatment on GD 2.5, 4.5, and 6.5 for the acute treatment. However, protective effects against fetal loss were not shown. Therefore, we applied bvPLA2 using a prophylactic method, meaning that when transferring this treatment to human, RPL patients will have to be treated with bvPLA2 before they become pregnant with probability to get infected.

Inflammatory cytokines, including TNF-α and IFN-γ, have been reported to play a major role in maintaining homeostasis between the fetal and maternal immune systems during pregnancy [27,28]. The expression of cytokines such as macrophages, NK cells, and Th1 cells reflects the status of multifarious immune cells, which target uterine endothelial cells to elicit vascular injury and placental ischemia [29,30]. The inflammatory cytokine TNF-α has been reported to act in gestational tissues to damage the placental blood supply and functions and cause fetal injury, leading ultimately to placental and fetal expulsion [31]. IFN-γ inhibits the secretion of GM-CSF from the uterine epithelium necessary for successful pregnancy. In the immunologically mediated spontaneous-abortion-prone CBA/J × DBA/2 mouse model, the expression levels of TNF-α and IFN-γ were significantly higher in the placentas of mice undergoing resorption compared with normal murine pregnancies [32]. It is well supported that the bacterial endotoxin LPS triggers macrophage-derived TNF-α release, which activates NK cells and IFN-γ secretion, resulting in an additive effect of TNF-α production. In our data, we showed elevated expression of TNF-α and IFN-γ in the serum samples of LPS-treated abortion mice, but bvPLA2 treatment dramatically decreased the serum inflammatory cytokine levels. The mRNA and protein levels of TNF-α expression were further analyzed in the uterine tissues. This suggests that the use of bvPLA2 as a Treg inducer to treat LPS-induced pregnancy loss succeeds by lowering the serum TNF-α and IFN-γ levels and uterine TNF-α secretion. Concerns about the correlation between Th1 type immunity and the bvPLA2-induced Treg population will be further elucidated.

Uterine NK cells are short-lived, terminally differentiated, and the most abundant decidual lymphocytes present at the maternal–fetal interface [33,34]. These cells secrete unique receptor cytokines that are essential for survival of the fetus in both mice and humans [35]. Women with RPL showed increased numbers of peripheral blood NK cells either prior to or during pregnancy compared with healthy non-pregnant or pregnant women [36,37]. LPS-treated pregnant mice exhibited a significant decrease in the number of uNK cells, as well as a significant increase in the number of uterine proinflammatory neutrophils [38]. Furthermore, NK cell deficiency during pregnancy resulted in structural abnormalities of decidual blood vessels and the placenta, as well as an increased rate of fetal growth retardation [39]. RPL and implantation failures are related to the dysfunction of NK cytotoxicity, gene expression, and cytokine production. In our results, the uNK cells were significantly reduced by the injection of LPS in the uterine tissues. The bvPLA2 pretreatment restored the uNK cell numbers to the pregnant levels. The cross-talk or relationship between uNK cells and Treg populations has yet to be elucidated. 

In a previous study, Shima et al. depleted Treg cells using an anti-CD25 antibody in allogeneic or syngeneic pregnant mice [40]. They found that the Treg depletion in early pregnancy (on GD 2.5) induced fetal loss in allogeneic pregnant mice but not in syngeneic pregnant females. Furthermore, depletion in the late stage of pregnancy (on GD 10.5 and 13.5) did not induce abnormalities such as fetal resorption or proteinuria. We also hypothesized the key role of Treg cells in bvPLA2-treated protection of LPS-induced abortion mice. A previous report relied primarily on anti-CD25 antibodies to deplete Treg populations. However, the use of anti-CD25 antibody was not specific to Treg populations because it also depletes CD25-expressing effector T cells, NK cells, and B cells [41]. To overcome this issue, we depleted Foxp3^+^ Treg cells by administering DT to DEREG mice carrying a Foxp3-DTR/eGFP transgene. The data showed that depletion of Tregs eliminated bvPLA2-mediated protection, including the fetal weight, viable implantation, and fetal resorption associated with LPS-treated fetal loss.

In conclusion, our data provide new evidence determining bvPLA2 as a therapeutic candidate in maternal–fetal immune tolerance through regulation of Treg cell differentiation and the expression of associated inflammatory factors. These findings will be informative for the development of treatment strategies for spontaneous abortion. 

## 4. Materials and Methods

### 4.1. Mice

Eight- to ten-week-old male and female C57BL/6 mice were obtained from Charles River Korea (OrientBio, Sungnam, Korea). Depletion of regulatory T cell (DEREG) mice (C57BL/6-Tg(Foxp3-DTR/EGFP)23.2Spar/Mmjax) were obtained from Jackson Laboratory (Bar Harbor, ME, USA). All animals were maintained in a barrier facility with air conditioning and a 12 h light and 12 h dark cycle. Mice had free access to food and water during the experiments. Animal Care and the Guiding Principles for Animal Experiment Using Animals was approved by the University of Kyung Hee Animal Care and Use Committee (KHUASP(SE)-18-002; 7 February, 2018). 

### 4.2. LPS-Induced Abortion Mouse Model

One male mouse and two female mice were housed together for mating. The female mice were inspected the next morning for the presence of a copulation plug, the presence of which was designated gestation day (GD) 0.5. The mated female mice were separated from the male mice. A dose of 0.4 µg/mouse of lipopolysaccharide (LPS from *Salmonella enteritidis*; Sigma, St. Louis, MO, USA) in 100 µL of phosphate-buffered saline (PBS) was injected intraperitoneally (i.p.) at GD 9.5. The negative control groups were treated with PBS alone. The LPS + bvPLA2 groups were treated with an i.p. injection of 0.5 mg/kg bvPLA2 (Sigma-Aldrich) once a week for two weeks prior to mating. Pregnant animals were sacrificed by cervical dislocation at GD 11.5. The implantation sites were opened longitudinally, and the contents of the uterus were examined for viable and resorbing embryos. The fetuses and placentas were weighed and inspected for malformations. The fetal resorption rates were calculated according to the following formula: number of resorptions/(number of viable fetus + resorptions). For Treg depletion experiments, DEREG mice were administered 1 µg of diphtheria toxin (DT; Sigma-Aldrich) diluted in PBS on −14, −13, −7, −6, −1, 3, and 7 days. DT was injected for two consecutive days 2 weeks before mating and again 3 and 7 days after mating. There were five to seven mice per group.

### 4.3. Flow Cytometric Analysis

Mononuclear cells from uterine tissues were aliquoted into tubes and were washed once in stain buffer (BD Biosciences, San Diego, CA, USA). For the analysis for various immune-related cells, allophycocyanin (APC)-conjugated anti-CD3 (clone 145-2C11; eBioscience, San Diego, CA, USA), fluorescein isothiocyanate (FITC)-conjugated anti-CD4 (clone GK1.5; eBioscience), phycoerythrin (PE)-labeled anti-CD8 (clone 53-6.7; eBioscience), rhodamine-conjugated agglutinin (from *Dolichos Biflorus*; Vector Laboratories, Buringame, CA, USA), and APC-conjugated anti-CD206 (clone C068C2; BioLegend, San Diego, CA, USA) antibodies were incubated on ice for 30 min. The stained cells were acquired using a FACSCalibur flow cytometer (BD Biosciences, San Jose, CA, USA), and the data were analyzed using FLOWJO software v10 (Tree star, Ashland, OR, USA). 

### 4.4. Measurement of Serum Cytokine Levels

Serum was obtained by the centrifugation of blood samples at 10,000*g* for 20 min. The levels of IL-4, IFN-γ, TNF-α, IL-17, and IL-10 in serum were determined using the Cytometric Bead Array (CBA) Mouse Th1/Th2/Th17 Cytokine Kit (BD Biosciences) according to the manufacturer’s protocols. Serum samples were added to a mixture of capture antibody bead reagent and PE-conjugated detection antibody and incubated at room temperature. After washing, the data were acquired using a BD FACS Calibur flow cytometer and were analyzed using BD CellQuest, BD CBA software (BD Biosciences). Individual cytokine concentrations were indicated by their fluorescence intensities. 

### 4.5. Real-Time PCR

Total RNA isolation from frozen decidual tissue was performed using an Easy-BLUE RNA extraction kit (iNtRON Biotechnology, Inc., Seongnam, Korea). Reverse transcription reactions were then carried out using CycleScript Reverse Transcriptase and Random Oligonucleotide Primers (Bioneer, Daejeon, Korea) following the manufacturer’s instructions. Quantitative real-time PCR was performed using the SensiFAST SYBR No-ROX kit (Bioline, Taunton, MA, USA) and analyzed using the LightCycler 480 system (Roche Ltd., Basel, Switzerland). All samples were run in duplicate, and the output level was reported as the average of the two cells. The PCR reactions were subjected to 55 cycles of denaturation at 95 °C for 10 s, annealing at 72 °C for 10 s, and extension at 60 °C for 10 s, with fluorescence measured at the end of each cycle. The mRNA levels were normalized to those of GAPDH. The primers used were as follows: TNF-α, forward: 5′-TTCTGTCTACTGAACTTCGGGGTGATCGGTCC-3′; reverse: 5′-GTATGAGATAGCAAATCGGCTGACGGTGTCCC-3′; GAPDH, forward: 5′-ACCCAGAAGACTGTGGATGG-3′; reverse: 5′-CACATTGGGGGTAGGAACAC-3′.

### 4.6. ELISA Detection for TNF-α

The concentration of TNF-α was measured using a quantitative sandwich enzyme-linked immunoassay kit (BD Biosciences) following the manufacturer’s instructions. Microtiter plates were incubated overnight at 4 °C with anti-mouse TNF-α monoclonal antibody in coating buffer. The wells were then washed with PBS containing 0.05% Tween-20 (Sigma-Aldrich) and were blocked with 5% FBS and 1% BSA in PBS. Subsequently, each well was loaded with 100 µL of BALF and was incubated for 2 h at RT. The wells were then incubated with biotinylated anti-mouse TNF-α monoclonal secondary antibody in assay diluents for 1 hr. Finally, the plates were treated with TMB substrate solution (KPL, San Diego, CA, USA) for 30 min, and the reaction was stopped by the addition of TMB stop solution. The optical density was measured at 450 nm in a microplate reader. The standard curve covered the range of 15.6 pg/mL to 1000 pg/mL (the detection limit). The TNF concentration was expressed as pg/mg protein.

### 4.7. Histology Analysis

For periodic acid–Schiff (PAS) staining, the sections were deparaffinized and rehydrated. The slides were placed in 1% periodic acid (Sigma-Aldrich) for 20 min at room temperature (RT) and then in Schiff’s reagent (Sigma-Aldrich) for 20 min. After rinsing twice for 5 min under tap water, the slides were counterstained with Harris’s hematoxylin (Sigma-Aldrich). For the detection of DBA lectin, the histological sections were deparaffinized, rehydrated, and heated in 10 mM sodium citrate buffer (pH 6.0) by autoclave and were further incubated for 20 min with 3% hydrogen peroxide (Sigma-Aldrich) to quench the endogenous peroxidase activity. After washing with PBS, nonspecific binding was reduced by blocking the sections with 1.5% bovine serum albumin (BSA) in PBS for 1 h. The sections were incubated with biotinylated DBA lectin (Vector Laboratories, Burlingame, CA, USA) overnight at 4 °C. After washing, the slides were incubated with an avidin–biotin complex kit (Vectastain ABC kit; Vector Laboratories). The slides were incubated in fresh 3,3′-diaminobenzidine-HCl (DAB) and rinsed under tap water. The labeled tissue sections were then mounted and analyzed under a bright-field microscope (Nikon, Tokyo, Japan). 

### 4.8. Statistical Analysis

The results were expressed as the arithmetic means of triplicates ± SEM. First, normality testing was performed using the D’Agostino–Pearson normality test. Significant differences were analyzed using one-way ANOVA followed by Tukey’s multiple comparison tests or Student’s *t*-tests using Prism 5.01 software (GraphPad Software Inc., San Diego, CA, USA). For non-normally distributed datasets, Kruskal–Wallis testing followed by Mann–Whitney testing was performed. Outliers were removed using the regression and outlier removal (ROUT) method with Q = 1 in GraphPad Prism. Differences with a *p*-value of <0.05 were considered statistically significant.

## Figures and Tables

**Figure 1 toxins-11-00404-f001:**
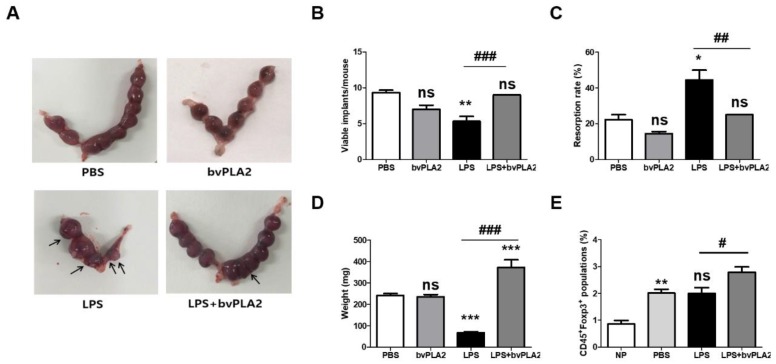
Effects of bee venom phospholipase A2 (bvPLA2) treatment on lipopolysaccharide (LPS)-induced fetal loss. One male mouse and two female mice were housed together for mating. The female mice were inspected the next morning, and the presence of vaginal plaque was designated gestation day (GD) 0.5. Next, 0.4 µg of LPS was injected intraperitoneally (i.p.) at GD 9.5. The LPS + bvPLA2 groups were treated with an i.p. injection of 0.5 mg/kg bvPLA2 once a week for two weeks prior to mating. Pregnant animals were sacrificed at GD 11.5 to investigate the pregnancy status. The fetuses and placentas were weighed and inspected for malformations. (**A**) Photographs of uterine horns represent the pregnancy status. The resorbed embryos were atrophied, and necrotic horns represent the pregnancy status (arrows). (**B**) Viable implants, (**C**) resorption rates, and (**D**) fetal weights are depicted. The fetal resorption rates were calculated as follows: number of resorptions/(number of viable fetuses + resorptions). (**E**) The percentages of CD45^+^Foxp3^+^ populations were determined from the uterine tissues. Significance: ^*^
*p* < 0.05, ^**^
*p* < 0.01, and ^***^
*p* < 0.001 vs. the PBS group; ^#^
*p* < 0.05, ^##^
*p* < 0.01, and ^###^
*p* < 0.001 vs. the LPS group; ns: no significance.

**Figure 2 toxins-11-00404-f002:**
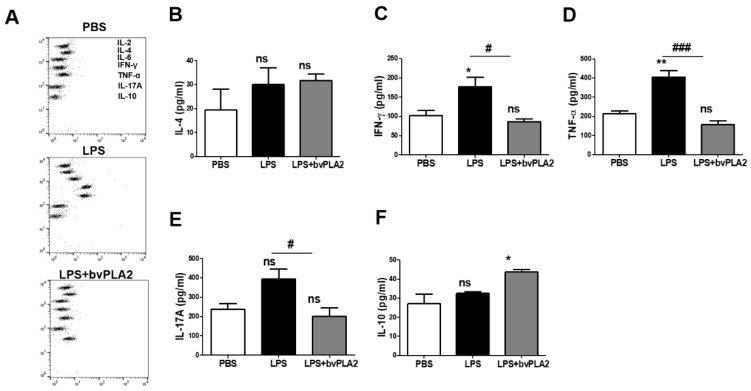
Effect of bvPLA2 treatment on serum cytokine levels in LPS-induced abortion mice. The blood samples of normal pregnant mice (PBS), abortion model mice (LPS), and bvPLA2-treated abortion model mice (LPS + bvPLA2) were collected for cytokine determination using the Cytometric Bead Array (CBA) Mouse Th1/2/17 Cytokine Kit. (**A**) The dots from top to bottom represent IL-2, IL-4, IL-6, IFN-γ, TNF-α, IL-17A, and IL-10, respectively. Concentrations of (**B**) IL-4, (**C**) IFN-γ, (**D**) TNF-α, (**E**) IL-17A, and (**F**) IL-10 in the serum of the PBS, LPS, and LPS + bvPLA2 groups are depicted. Significance: ^*^
*p* < 0.05, ^**^
*p* < 0.01 vs. the PBS group; ^#^
*p* < 0.05, ^##^
*p* < 0.01, and ^###^
*p* < 0.001 vs. the LPS group; ns: no significance.

**Figure 3 toxins-11-00404-f003:**
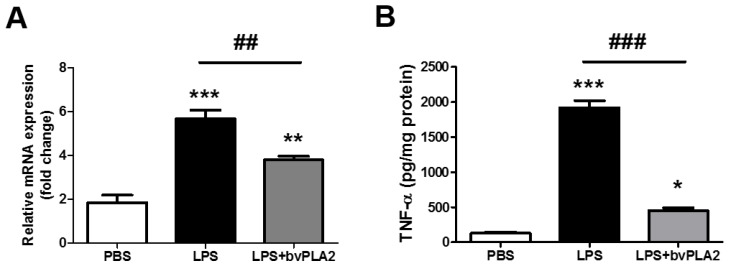
Effects of bvPLA2 treatment on TNF-α expression in the uterine tissues in LPS-induced abortion mice. (**A**) TNF-α mRNA expression in the uterine tissue was quantified by real-time PCR. (**B**) The concentrations of TNF-α in the uterine tissue were determined by ELISA. Significance: ^*^
*p* < 0.05, ^**^
*p* < 0.01 and ^***^
*p* < 0.001 vs. the PBS group; ^##^
*p* < 0.01 and ^###^
*p* < 0.001 vs. the LPS group.

**Figure 4 toxins-11-00404-f004:**
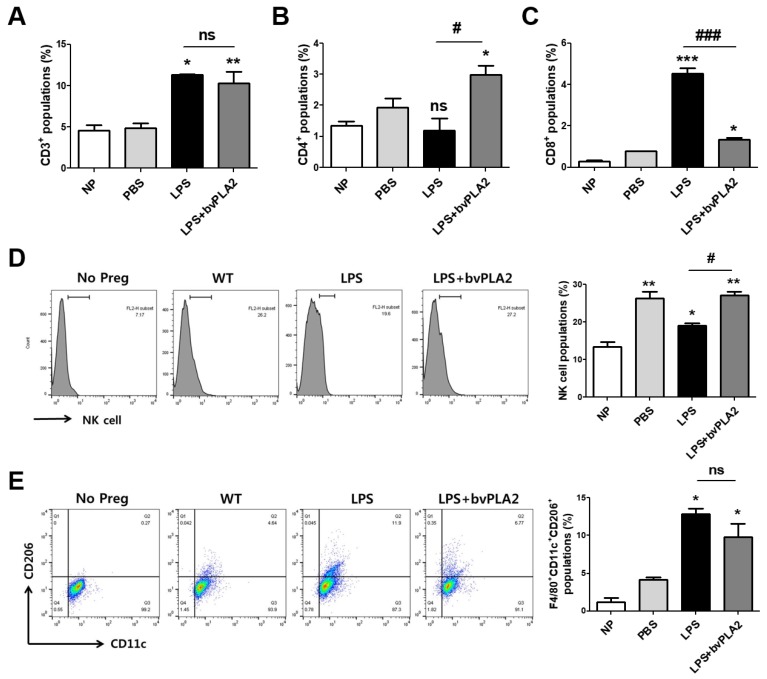
Modulation of immune cell types in LPS-induced abortion mice exposed to bvPLA2. Monocytes were isolated from the uterine tissues of female mice on GD 11.5 after treatment with PBS, LPS, and LPS + bvPLA2 or from non-pregnant females. The percentages of (**A**) CD3^+^ total T cells, (**B**) CD4^+^ T helper cells, and (**C**) CD8^+^ cytotoxic T cells from the uterine tissues are depicted. (**D**) Representative histogram and the percentage of DBA-lectin^+^ uNK cells stained and analyzed by flow cytometry. (**E**) Dot plot and the percentage of F4/80^+^CD11c^+^CD206^+^ M2 macrophages stained and analyzed by flow cytometry. Significance: ^*^
*p* < 0.05, ^**^
*p* < 0.01, and ^***^
*p* < 0.001 vs. the NP group; ^#^
*p* < 0.05 and ^###^
*p* < 0.001 vs. the LPS group; ns: no significance.

**Figure 5 toxins-11-00404-f005:**
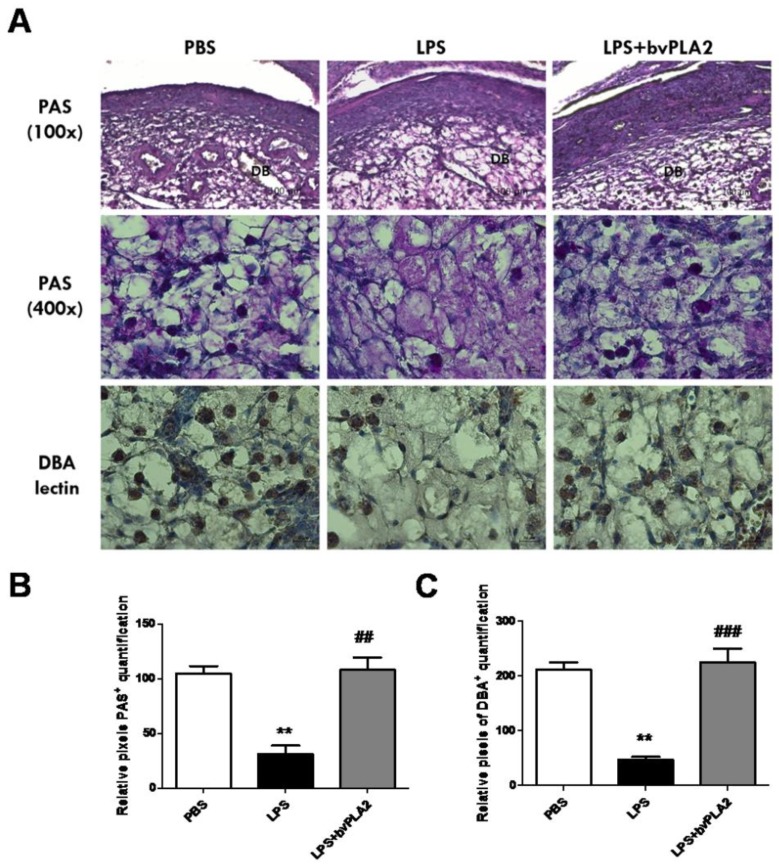
Alteration of uterine natural killer (uNK) cells in the mouse uterus exposed to bvPLA2. (**A**) uNK cells were determined by periodic acid–Schiff (PAS) staining (100× upper and 400× lower) and immunohistochemical staining with biotinylated *Dolichos biflorus* agglutinin (DBA) lectin in the uterine tissues. Scale Bars: 100 μm for PAS (100×) panel and 20 μm for PAS (400×) panel. The figure represents sections from five individual mice. DB: decidua basalis. Bar graphs of data pertaining to the effects of bvPLA2 pretreatment on uNK cells by (**B**) PAS staining and (**C**) DBA lectin. Significance: ^**^
*p* < 0.01 vs. the PBS group; ^##^
*p* < 0.01 and ^###^
*p* < 0.001 vs. the LPS group.

**Figure 6 toxins-11-00404-f006:**
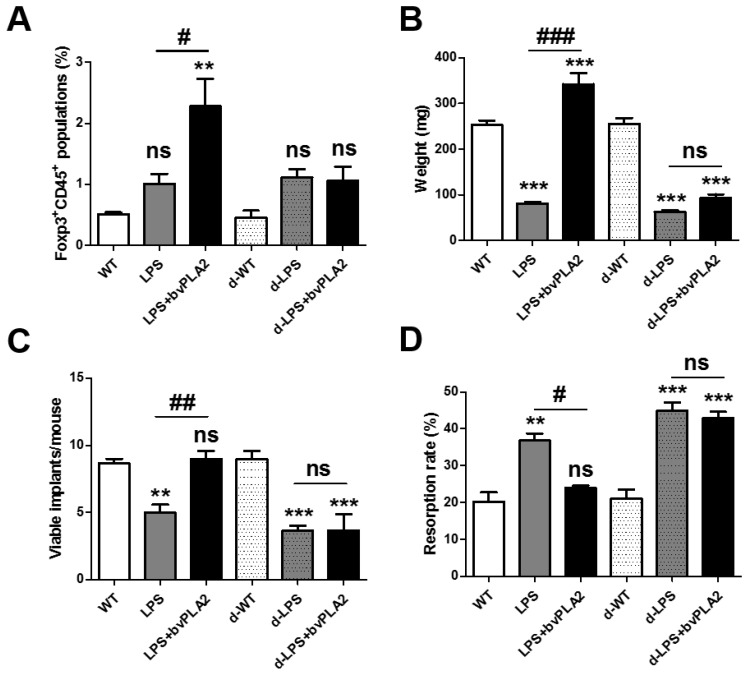
Effects of bvPLA2 on fetal loss in Treg-depleted abortion-prone mice. Monocytes were isolated from the uteruses of pregnant female mice on GD 11.5 after treatment with PBS, LPS, or LPS + bvPLA2 with or without DT administration. (**A**) Day 5 after DT injection, splenocytes were isolated and CD45^+^Foxp3^+^ Treg cells were analyzed by FACS. (**B**) Fetal weight, (**C**) viable implants, and (**D**) resorption rates are depicted. Significance: ^**^
*p* < 0.01 and ^***^
*p* < 0.001 vs. the PBS group; ^#^
*p* < 0.05, ^##^
*p* < 0.01, and ^###^
*p* < 0.001 vs. the LPS group; ns: no significance.

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
