# Peer review of "Prophylactic Effects of Bee Venom Phospholipase A2 in Lipopolysaccharide-Induced Pregnancy Loss"

_toxins, 2019, doi:10.3390/toxins11070404_

Round 1
Reviewer 1 Report
The manuscript is properly revised according to my comments and has been significantly improved by revising Fig. 1 and the result section 2.1.

Author Response
Reviewer 1
I review the manuscript [toxins-542859] entitled Prophylactic effects of bee venom phospholipase A2 in lipopolysaccharide-induced pregnancy loss.
As the manuscript is properly revised according to my comments and has been significantly improved by revising Fig. 1 and the result section 2.1, it is acceptable.
Response: We thank for the reviewer’s comments
Reviewer 2 Report
In this revised manuscript, the authors have included the controls for bvPLA2-only treatment in Figure 1. This strongly suggests a real effect on fetal weight (Fig 1D and Fig 6B) from bvPLA2+LPS treatment. This enhancement effect was also noticeable in IL-10 levels (Fig 2F) and Foxp3+CD45+ populations (Fig 6A). Although the mechanism is not clear from the available data, the authors should point this out in their discussion and suggest possible connection to the promotion of Treg by bvPLA2.
The authors has indicated in their response to the reviewer, "In addition, we added the mentions to provide possible explanations for the apparent enhancements with bvPLA2+LPS in Figure 1D and 1E as “We suggest that bvPLA2-mediated induction of Foxp3+Treg cells in uterine tissues may affect to the prevention of pregnancy loss." It should be noted that Figure 1E has been removed from the current revision. The "explanations for the apparent enhancements with bvPLA2+LPS" seemed less than obvious. The data of previous Fig 1E should be described in the manuscript even without the bvPLA2-only treatment control. The "enhancement with bvPLA2+LPS" should be clearly stated in the discussion.
There are additional minor points that the authors should correct in their final manuscript.
"DEREG mouse model" should be added to the key words list.
line 71, bVPLA2 should be corrected as bvPLA2.
line 89-94, the unrealistic precision of the values should be corrected as 44.4±9.6% ,
25%, 22.2±4.8, 67±15%, 242±28 mg, 67±15 mg and 373±105 mg.
Dosages for DT injection of DEREG mice should be described in section 2.6 .
Author Response
Reviewer 2
In this revised manuscript, the authors have included the controls for bvPLA2-only treatment in Figure 1. This strongly suggests a real effect on fetal weight (Fig 1D and Fig 6B) from bvPLA2+LPS treatment. This enhancement effect was also noticeable in IL-10 levels (Fig 2F) and Foxp3+CD45+ populations (Fig 6A). Although the mechanism is not clear from the available data, the authors should point this out in their discussion and suggest possible connection to the promotion of Treg by bvPLA2.
The authors has indicated in their response to the reviewer, "In addition, we added the mentions to provide possible explanations for the apparent enhancements with bvPLA2+LPS in Figure 1D and 1E as “We suggest that bvPLA2-mediated induction of Foxp3+Treg cells in uterine tissues may affect to the prevention of pregnancy loss." It should be noted that Figure 1E has been removed from the current revision. The "explanations for the apparent enhancements with bvPLA2+LPS" seemed less than obvious. The data of previous Fig 1E should be described in the manuscript even without the bvPLA2-only treatment control. The "enhancement with bvPLA2+LPS" should be clearly stated in the discussion.
Response: We thank for the reviewer’s comment. We added the explanations that bvPLA2-mediated induction of Foxp3+ Treg cells in uterine tissues may affect to the prevention of pregnancy loss. And also we inserted the figure showed the induction of Treg populations in the bvPLA2 treated uterine tissues in Figure 1E. (Page 3)
There are additional minor points that the authors should correct in their final manuscript.
"DEREG mouse model" should be added to the key words list.
Response: We added DEREG mouse model in the key words list (Page 1)
line 71, bVPLA2 should be corrected as bvPLA2.
Response: We apologize the mistake. Now we corrected bVPLA2 as bvPLA2 (Page 2)
line 89-94, the unrealistic precision of the values should be corrected as 44.4±9.6% ,
25%, 22.2±4.8, 67±15%, 242±28 mg, 67±15 mg and 373±105 mg.
Response: According to the reviewer’s comment, we corrected the unrealistic precision of the values (Page 2 and 3).
Dosages for DT injection of DEREG mice should be described in section 2.6 .
Response: We added the dosage for DT injection of DEREG mice in section 2.6 (Page 7)
Reviewer 3 Report
In this manuscript, the authors investigated the prophylactic effects of bee venom phospholipase A2 (bvPLA2) in a mouse model of lipopolysaccharide (LPS)-induced pregnancy loss.
Spontaneous abortion represents a common form of embryonic loss caused by early pregnancy failure, and it has been previously reported that a deficiency in the regulatory T cells (Treg) cell number and function is associated with unexplained infertility, abortion, and preeclampsia.
In the study presented here, the authors hypothesized that pre-treatment with bvPLA2, a known inducer of Treg, may reduce the abortion rate in a mouse model of LPS-induced pregnancy loss.
When mice were pre-treated with bvPLA2, a reduction of fetal loss was observed, compared to controls. Treg cells were significantly increased, compared with those in the non-pregnant, PBS and LPS groups. After LPS injection, the levels of proinflammatory cytokines were markedly increased compared with those in the PBS mouse group, while bvPLA2 treatment showed significantly decreased TNF-α and IFN-γ expression compared with that in the LPS group
The authors’ results let them to propose bvPLA2 as a novel therapeutic candidate to support the maternal-fetal immune tolerance through regulation of Treg cells differentiation and expression of associated inflammatory factors.
Overall, this review is well written, the experimental setup looks adequate, and presents some interesting points for a discussion; I think it could be of interest to the broad readership of Toxins.
Minor points:
Fonts accompanying x and y axes in Figures 2 and 4 are too small
There are some misspelled words/odd sentences through the text. One example on line 232: “is induces secretary changes…” I suppose the correct sentence is “induces secretory changes…”)
Author Response
Reviewer 3
In this manuscript, the authors investigated the prophylactic effects of bee venom phospholipase A2 (bvPLA2) in a mouse model of lipopolysaccharide (LPS)-induced pregnancy loss.
Spontaneous abortion represents a common form of embryonic loss caused by early pregnancy failure, and it has been previously reported that a deficiency in the regulatory T cells (Treg) cell number and function is associated with unexplained infertility, abortion, and preeclampsia.
In the study presented here, the authors hypothesized that pre-treatment with bvPLA2, a known inducer of Treg, may reduce the abortion rate in a mouse model of LPS-induced pregnancy loss.
When mice were pre-treated with bvPLA2, a reduction of fetal loss was observed, compared to controls. Treg cells were significantly increased, compared with those in the non-pregnant, PBS and LPS groups. After LPS injection, the levels of proinflammatory cytokines were markedly increased compared with those in the PBS mouse group, while bvPLA2 treatment showed significantly decreased TNF-α and IFN-γ expression compared with that in the LPS group
The authors’ results let them to propose bvPLA2 as a novel therapeutic candidate to support the maternal-fetal immune tolerance through regulation of Treg cells differentiation and expression of associated inflammatory factors.
Overall, this review is well written, the experimental setup looks adequate, and presents some interesting points for a discussion; I think it could be of interest to the broad readership of Toxins.
Minor points:
Fonts accompanying x and y axes in Figures 2 and 4 are too small
Response: We adjusted the fonts in Figure 2 and 4.
There are some misspelled words/odd sentences through the text. One example on line 232: “is induces secretary changes…” I suppose the correct sentence is “induces secretory changes…”)
Response: We carefully re-checked the word/odd sentences and corrected the misspelled sentences (Page 6, 8, and 10).
This manuscript is a resubmission of an earlier submission. The following is a list of the peer review reports and author responses from that submission.
Round 1
Reviewer 1 Report
1. General comments
In the manuscript, it is reported a new evidence that bvPLA2 was a therapeutic candidate in maternal-fetal immune tolerance by regulation of Treg cell differentiation and expression of associated inflammatory cytokines. The findings will be helpful for the development of therapeutic strategies for spontaneous abortion.
2. Minor points
It is necessary to check or correct the following points.
1) Legend of Figure 1
Correct “(B) Viable implants, (C) resorption rates, and fetal weights are depicted” to “(B) Viable implants, (C) resorption rates, and (D) fetal weights are depicted”.
Correct “(D) The percentages of ---” to “(E) The percentages of ---”.
2) Lane 95~96
Correct the sentence “---, we examined the levels in reproductive tissues” to “---, we examined the levels in reproductive tissues (Fig. 1E)".
It is recommended to correct “---was significantly increased after pregnancy---” to “---was significantly increased after pregnancy (the PBS group)---”.
3) Lane 199~200: Correct the sentence “---abortion-prone mice (the d-LPS+bvPLA2 group) (Fig. 6B)” to “---abortion-prone mice (the d-LPS+bvPLA2 group) (Fig. 6A)”.
4) Lane 204: Correct the word “(Fig. 6C)” to “(Fig. 6B)”.
5) Lane 206: Correct the word “(Fig. 6D)” to “(Fig. 6C)”.
6) Lane 209: Correct the sentence “Treg depletion (Fig. 6D and 6E)” to “Treg depletion (Fig. 6C and 6D)”.
7) Legend of Figure 6: It is essential to check the figure legend, B) ~ E).
8) Lane 233-234: Correct the sentence “is induces” to “induces”.
9) Lane 242: Correct the sentence “fetal loss y bvPLA2” to “fetal loss by bvPLA2”.
10) Lane 372-373: Correct the sentence “by the addition stop solution” to “by the addition of stop solution”.

Author Response
Dear Toxins editor:
Thank you the reviews of our manuscript “Protective effects of bee venom phospholipase A2 in lipopolysaccharide-induced pregnancy loss”. We thank the reviewer and editor for their positive comments and suggestions. We have addressed the comments of reviewer as described below. Changes in the text are highlighted in the revised manuscript.
Reviewer #1
1. General comments
In the manuscript, it is reported a new evidence that bvPLA2 was a therapeutic candidate in maternal-fetal immune tolerance by regulation of Treg cell differentiation and expression of associated inflammatory cytokines. The findings will be helpful for the development of therapeutic strategies for spontaneous abortion.
Response: Thank you for the comments
2. Minor points
It is necessary to check or correct the following points.
1) Legend of Figure 1
Correct “(B) Viable implants, (C) resorption rates, and fetal weights are depicted” to “(B) Viable implants, (C) resorption rates, and (D) fetal weights are depicted”.
Correct “(D) The percentages of ---” to “(E) The percentages of ---”.
Response: We apologize for the mistake. Now we revised as the reviewer’s comments, as follows
“(B) Viable implants, (C) resorption rates, and (D) fetal weights are depicted” and “(E) The percentages of …” (Page 3).
2) Lane 95~96
Correct the sentence “---, we examined the levels in reproductive tissues” to “---, we examined the levels in reproductive tissues (Fig. 1E)".
It is recommended to correct “---was significantly increased after pregnancy---” to “---was significantly increased after pregnancy (the PBS group)---”.
Response: We revised above sentences, as reviewer’s comment, as follows.
“..we examined the levels in reproductive tissues (Fig. 1E).” and (..was significantly increased after pregnancy (the PBS group) compared with that in non-pregnant mice (the NP group).” (Page 3)
3) Lane 199~200: Correct the sentence “---abortion-prone mice (the d-LPS+bvPLA2 group) (Fig. 6B)” to “---abortion-prone mice (the d-LPS+bvPLA2 group) (Fig. 6A)”.
Response: We apologize the mistake. Now we revised “Fig. 6B” to “Fig. 6A” (Page 6)
4) Lane 204: Correct the word “(Fig. 6C)” to “(Fig. 6B)”.
Response: We apologize the mistake. Now we revised “Fig. 6C” to “Fig. 6B” (Page 6)
5) Lane 206: Correct the word “(Fig. 6D)” to “(Fig. 6C)”.
Response: We apologize the repeated mistake. We revised “Fig. 6D” to Fig. 6C” (Page 6)
6) Lane 209: Correct the sentence “Treg depletion (Fig. 6D and 6E)” to “Treg depletion (Fig. 6C and 6D)”.
Response: We apologize the repeated mistake. We revised “Fig. 6D and 6E” to Fig. 6C and 6D” (Page 7)
7) Legend of Figure 6: It is essential to check the figure legend, B) ~ E).
Response: We corrected the figure legend, as follows.
“(B) Fetal weight, (C) viable implants, and (D) resorption rates are depicted.” (Page 7)
8) Lane 233-234: Correct the sentence “is induces” to “induces”.
Response: We revised the sentence, as reviewer’s comment.
“ Progesterone, an essential hormone in the process of reproduction, induces secretary changes in the inner lining…” (Page 8)
9) Lane 242: Correct the sentence “fetal loss y bvPLA2” to “fetal loss by bvPLA2”.
Response: We thank for the reviewer’s comment. We added the missing “b” (Page 8)
10) Lane 372-373: Correct the sentence “by the addition stop solution” to “by the addition of stop solution”.
Response: We revised the sentence, as reviewer’s comment.
“the reaction was stopped by the addition of stop solution.” (Page 10)
Reviewer 2 Report
In this manuscript, the authors reported their findings on protective benefit of pretreating bvPLA2 in blocking LSP-induced pregnancy failure. The authors hypothesized the action of bvPLA2 is through its promotion of Treg, that subsequently allows the mother to fight off the effect of LPS. Their finding has great potential for elucidating the the immune modulation process during pregnancy for maintenance of pregnancy and protection of the fetus. It however has several major weaknesses.
The choice of the term “Spontaneous Abortion” in the title is misleading. In their mice model, they only observed fetal resorption not abortion. The authors should use “Pregnancy Loss” to characterize their study.
The authors did not include bvPLA2 treatment only controls. In the absence of this control group, it complicated the explanation why in the bvPLA2+LPS group, there appeared to have benefits over the PBS group in several measurements, including fetus weight, anti-inflammatory cytokines, and protective T-cell populations. The authors did not make any attempt to explain these potentially significant findings.
There are major errors in several figure legends. For Fig 1, there are 5 panels, A-E, but only 4 items were explained, A-D. The explanations for these four did not match what were shown in the figure. For Fig 2 and Fig 4, the figures contained both flow cytometry/immuno multi-plex data and bar graphs. It is difficult for the readers to decipher the graph. More clear explanations should be provided in the legends . For Fig 5, the descriptions A, B, C and DB were not even shown in the graph. A reference bar/scale should be provided for the micro photograph panels. The images of the two different stains are not likely from the same samples, it should be clearly stated. For Fig 6, there are 4 panels with descriptions for 5 items. It also is not clear if the day1 day5 data are corresponding to WT and d-WT in the graph. If they are, it should be made clear in the figure legend. In that case, what was the result for untreated cells?
Author Response
Dear Toxins editor:
Thank you the reviews of our manuscript “Protective effects of bee venom phospholipase A2 in lipopolysaccharide-induced pregnancy loss”. We thank the reviewer and editor for their positive comments and suggestions. We have addressed the comments of reviewer as described below. Changes in the text are highlighted in the revised manuscript.
Reviewer #2
In this manuscript, the authors reported their findings on protective benefit of pretreating bvPLA2 in blocking LSP-induced pregnancy failure. The authors hypothesized the action of bvPLA2 is through its promotion of Treg, that subsequently allows the mother to fight off the effect of LPS. Their finding has great potential for elucidating the the immune modulation process during pregnancy for maintenance of pregnancy and protection of the fetus. It however has several major weaknesses.
The choice of the term “Spontaneous Abortion” in the title is misleading. In their mice model, they only observed fetal resorption not abortion. The authors should use “Pregnancy Loss” to characterize their study.
Response: We thank for the reviewer’s comments. As reviewer’s comment, we revised the title as follows,
“Prophylactic effects of bee venom phospholipase A2 in lipopolysaccharide-induced pregnancy loss”
The authors did not include bvPLA2 treatment only controls. In the absence of this control group, it complicated the explanation why in the bvPLA2+LPS group, there appeared to have benefits over the PBS group in several measurements, including fetus weight, anti-inflammatory cytokines, and protective T-cell populations. The authors did not make any attempt to explain these potentially significant findings.
Response: We thank for the reviewer’s comment. We did not determine the effects of bvPLA2 only treatment. We are planning to investigate the effects of bvPLA2 treatment only control in the further study. We added the explanation about the bvPLA2 treatment and protective effects in fetal loss in discussion section (page 8).
There are major errors in several figure legends. For Fig 1, there are 5 panels, A-E, but only 4 items were explained, A-D. The explanations for these four did not match what were shown in the figure.
Response: We thank for the comment. We corrected the errors in figure legend for Fig. 1 and added the explanation for Fig. 1E.
“we examined the levels in reproductive tissues (Fig. 1E).” (Page 3)
“(A) Photographs of uterine horns represent the pregnancy status. The resorbed embryos were atrophied, and necrotic horns represent the pregnancy status (arrows). (B) Viable implants, (C) resorption rates, and (D) fetal weight are depicted. The fetal resorption rates were calculated as follows: number of resorptions/(number of viable fetuses + resorptions). (E) The percentages of the CD45+Foxp3+ populations were determined from the uterine tissues” (Page 3)
For Fig 2 and Fig 4, the figures contained both flow cytometry/immuno multi-plex data and bar graphs. It is difficult for the readers to decipher the graph. More clear explanations should be provided in the legends.
Response: We thank for the comments. We added more clear explanations in the legends.
“Concentrations of (B) IL-4, (C) IFN-r, (D) TNF-a, (E) IL-17A, and (F) IL-10 in the serum of the PBS, LPS, and LPS+bvPLA2 group were depicted.” (Page 4, for Fig. 2)
“The percentages of (A) CD3+ total T cells, (B) CD4+ T helper cells, and (C) CD8+ cytotoxic T cells were from the uterine tissues were depicted. (D) Representative histogram and the percentage of DBA-lectin+ uNK cells were stained and analyzed by flow cytometry. (E) Dot plot and the percentage of F4/80+CD11cc+CD206+ M2 macrophages were analyzed by FACS.” (Page 5 and 6, for Fig. 4)
For Fig 5, the descriptions A, B, C and DB were not even shown in the graph. A reference bar/scale should be provided for the micro photograph panels. The images of the two different stains are not likely from the same samples, it should be clearly stated.
Response: We thank for the comment. We revised the correct legend for Fig. 5, as follows.
“ uNK cells were determined by PAS staining (upper panel) and immunohistochemical staining with biotinylated DBA lectin (lower panel) in the uterine tissues (Magnification = 400x). The figure represents sections from five individual mice.” (Page 6)
For Fig 6, there are 4 panels with descriptions for 5 items. It also is not clear if the day1 day5 data are corresponding to WT and d-WT in the graph. If they are, it should be made clear in the figure legend. In that case, what was the result for untreated cells?
Response: We thank for the comments. We corrected the figure legend, as follows,
“(A) Day 5 after DT injection, splenocytes were isolated and …(B) Fetal weight, (C) viable implants, and (D) resorption rated are depicted.” (Page 7)
Round 2
Reviewer 2 Report
The authors have corrected most errors in their previous manuscript.
They also responded in communication on the previous comments:
“The authors did not include bvPLA2 treatment only controls. In the absence of this control group, it complicated the explanation why in the bvPLA2+LPS group, there appeared to have benefits over the PBS group in several measurements, including fetus weight, anti-inflammatory cytokines, and protective T-cell populations. The authors did not make any attempt to explain these potentially significant findings. “
They did not address this issue in their revised manuscript. The best solution is to provide direct evidence using bvPLA2 controls. At least the authors should clearly point this out in the manuscript and provide possible explanations for the apparent enhancements with bvPLA2+LPS in Figure 1D and 1E.
Author Response
Reply to reviewer's comments,
They also responded in communication on the previous comments:
“The authors did not include bvPLA2 treatment only controls. In the absence of this control group, it complicated the explanation why in the bvPLA2+LPS group, there appeared to have benefits over the PBS group in several measurements, including fetus weight, anti-inflammatory cytokines, and protective T-cell populations. The authors did not make any attempt to explain these potentially significant findings. “
They did not address this issue in their revised manuscript. The best solution is to provide direct evidence using bvPLA2 controls. At least the authors should clearly point this out in the manuscript and provide possible explanations for the apparent enhancements with bvPLA2+LPS in Figure 1D and 1E.
Response: We thank for the reviewer’s comment. We did not determine the effects of bvPLA2 only treatment. We are asking a favor not performing this experiment in this manuscript, because of time limitation and difficulties in approving redundant protocols from animal ethnic committee. However, we are planning to investigate the effects of bvPLA2 treatment only control in the future study. We added the explanation about the bvPLA2 treatment and protective effects in fetal loss in discussion section as “We suggest that the induction of Treg population by bvPLA2 treatment resulted to prevent the fetal loss and increase protective T cell populations”. In addition, we added the mentions to provide possible explanations for the apparent enhancements with bvPLA2+LPS in Figure 1D and 1E as “We suggest that bvPLA2-mediated induction of Foxp3+Treg cells in uterine tissues may affect to the prevention of pregnancy loss. We further investigate the effects of bvPLA2 on the inflammatory cytokine levels, including IFN-γ and TNF-α, in uterine tissues.”
